# On the Exceptionally High Loading of L-Proline on Multi-Wall Carbon Nanotubes

**Jiafang Xu, Jichao Liang, Sheng Huang, Ge Yang, Keyi Tian, Ruonan Chen, Hongyu Chen and Yanhua Zhang \***

Institute of Advanced Synthesis and School of Chemistry and Molecular Engineering, Nanjing Tech University, Nanjing 211816, China; jiafang_xu0611@163.com (J.X.); ljc1997@njtech.edu.cn (J.L.); ias_huangsheng@njtech.edu.cn (S.H.); yangge05@njtech.edu.cn (G.Y.); tiankeyi@njtech.edu.cn (K.T.); rnchen@njtech.edu.cn (R.C.); iashychen@njtech.edu.cn (H.C.)

\* Correspondence: ias_yhzhang@njtech.edu.cn; Tel.: +86-025-58139046

**Abstract:** L-proline is directly loaded on the multi-wall carbon nanotubes (MWCNTs) with exceptionally high loading content of 67 wt.%. The obtained L-proline/MWCNTs catalyst is on par with the catalytic activity of free L-proline, even after 7 rounds of recycling and reusing. The excellent activity of L-proline/MWCNTs in typical Aldol reaction, Mannich reaction, Michael reaction, $\alpha$-oxyamination reaction, and Knoevenagel condensation shows a broad applicability of the composite catalyst in different reactions and solvent systems. We believe that the unusual loading mode may open a window for designing heterogenized organo-catalysts.

**Keywords:** L-proline/MWCNTs; high loading; recycling stability; catalytic efficiency; heterogenized organo-catalysis

## 1. Introduction

Asymmetric reactions catalyzed by small organic molecules (i.e., organo-catalysts) have the advantages of high enantioselectivity under mild reaction conditions [1,2]. However, the catalyst loading is usually high (~20 mol% or more) and the recovery of the catalyst is difficult [3,4]. As a result, the practical applications of organo-catalysts, especially in industrial setting, are greatly restricted [5,6].

The primary approach of recovering and reusing organo-catalysts is to fix them onto suitable substrates, to facilitate post-reaction treatment and product separation [7–10]. However, unlike inorganics that can form nanoparticles with large surface area, organic molecules are usually covalently tethered to the substrate surface and thus, the loading is greatly limited by the available functional groups [11,12]. Also, although the loaded organo-catalysts can be recycled and reused, their reactivity and selectivity are often significantly lower than that of the free organic catalysts [13]. This may be due to the heterogeneity of the homogeneous organic catalyst.

L-proline is a model organo-catalyst widely used in the heterogenization studies [14,15]. Many efforts have been made to recycle L-proline through fixation on appropriate substrates, such as polymers [16–18], oxides [19,20], and magnetic particles [21,22]. Typical studies focus on the proof-of-concept demonstration of recycling catalyst, using overall yield as the indicator of the catalytic activity. From the mode of catalyst heterogenization, it is obvious that the amount of L-proline would be much less than the underlying substrate, i.e., the polymer or inorganic substrates usually contain many layers of atoms, in contrast to the sporadic L-proline molecules tethered on their surface. The tethered organo-catalysts are known to have decreased catalytic activity, likely owing to their restricted conformations. For example, Chandrasekarna et al., report that L-proline could be covalently fixed on the MCM-41 (a silicon-based support), but the loading amount is only about 28 wt.%, and the catalytic efficiency is poor [23].

Carbon nanotubes (CNTs) are common catalyst support, as they possess superior properties of high aspect ratio, high chemical stability, high electrical and thermal conductivity [24,25]. However, its seamless tubular structure offers few functional groups for tethering, and extensive breaking of the C-C bonds would cause its disintegration [26]. Although the limited loading content is a minor issue for the efficient catalysts, it is non-ideal for the organo-catalysts. For example, Hajipour reported the loading of L-proline on the multi-wall carbon nanotubes (MWCNTs) through amidation reaction with the amino group on the functionalized MWCNTs. The content of L-proline in the obtained catalyst is only about 10 wt.% (i.e., 1.27 wt.% of N) [27]. Tagmatarchis [28] and Eshghi [29] reported the covalent functionalization of the MWCNTs with the proline derivatives, giving the maximum loading of 15 wt.% and 7 wt.%, respectively. When the catalysts are applied in Aldol reaction or Mannich reaction, although the yield of the product is up to 95%, the catalytic efficiency decreases dramatically after only 3 or 5 rounds of recycling.

Herein, we report a surprising finding that L-proline could be fixed onto the MWCNTs with exceptionally high loading content of 67 wt.%. Extensive control experiments show that L-proline is adsorbed on both inner and outer surface of the MWCNTs, and its loading content is confirmed by the thermogravimetric analysis (TGA), elemental analysis, and catalytic performance. The catalytic efficiency and recycling capacity of the obtained L-proline/MWCNTs catalyst are investigated in the typical catalytic reactions, including Aldol reaction [30,31], Mannich reaction [32], Michael reaction [33], $\alpha$-oxyamination reaction [34,35], and Knoevenagel condensation [36,37].

## 2. Results and Discussion

### 2.1. Characterization of the L-Proline/MWCNTs Catalysts

Figure 1 shows the TGA of the product and the related materials. MWCNTs (Figure 1(d1)) by themselves shows little loss of weight up to 800 °C, likely owing to the pyrolysis of the oxygen-containing functional groups. Neat L-proline (Figure 1(d2)) shows a characteristic weight-loss trace from 229 °C, decreasing to nearly zero weight at 347 °C. The L-proline/MWCNTs catalyst (Figure 1(d3)) shows a slight weight loss (11 wt.%) around 200 °C. However, the most striking aspect of the trace is the high percentage of weight loss that is similar to the L-proline trace. It is calculated that the content of L-proline in the catalyst is as high as 67 wt.%. Elemental analysis of the catalyst shows approximate 8.17 wt.% of N, which is equivalent to 67.3 wt.% of L-proline (5.84 mmol/g, Table S1, Supplementary Materials (SM)), matching with the TGA data.

The intuitive explanation for such extraordinary loading content is that it is possibly caused by the co-precipitation of L-proline with the MWCNTs. Considering the high solubility of L-proline in water (162 g in 100 g water) and the absence of the precipitant; however, it is inconceivable that the excess L-proline would be excluded from water in the first place and remain with the MWCNTs after the repeated washing cycles.

The initial 11 wt.% weight loss of the L-proline/MWCNTs catalyst in TGA is also mysterious, since there is no other chemical present during the catalyst loading. Our only guess is that it is due to the trapped water molecules. As water is not expected to get trapped in the absence of L-proline (Figure 1(d1)), its presence would indicate the adsorption of L-proline in the inner cavities of the MWCNTs as plugs. Actually, elemental analysis of the catalyst shows approximate 27.4 wt.% of O (Table S1, SM), much more than the oxygen contribution from L-proline. The difference of oxygen content would account for 12 wt.% of water, almost matching the initial 11 wt.% weight loss in TGA.

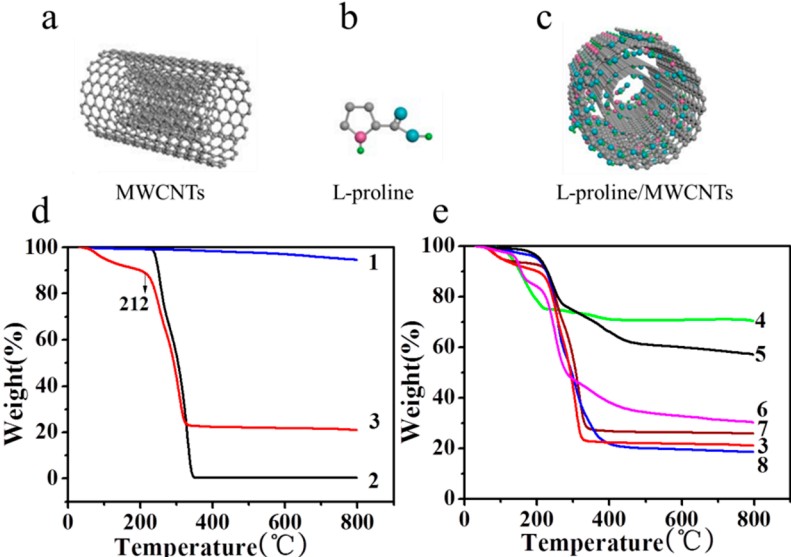

**Figure 1.** (**a**). MWCNTs, (**b**). neat L-proline, (**c**). L-proline/MWCNTs, (**d**). TGA of (1) MWCNTs, (2) neat L-proline, and (3) L-proline/MWCNTs catalyst after 3 rounds of washing with $H_2O$, (**e**). TGA of (4) L-proline/MWCNTs catalyst after additional washing with 10% aq. NaOH, (5) L-proline/SWCNTs catalyst, (6) L-proline/graphene oxide (GO) catalyst after 5 rounds of washing with $H_2O$, (7) L-proline/MWCNTs catalyst after 5 rounds of washing with $H_2O$, and (8) L-proline/MWCNTs catalyst after additional washing with 10% aq. $H_2SO_4$.

Single-wall CNTs are also good supports for L-proline, but the loading content is only 24 wt.% (Figure 1(e5)), far less than that of the MWCNTs. This comparison supports the loading of L-proline in the cavities of MWCNTs, considering that the latter has layers of inner space. Figure 1(e6) shows the trace of L-proline-loaded graphene oxide (GO), from which 38 wt.% loading content could be estimated. This result is consistent with the literature report (4.06 mmol/g, ~46 wt.%) [38]. In contrast, the loading content on activated carbon is only 3 wt.% (Figure 2(3)), much lower than the CNTs and GO, suggesting that the aromatic carbon may be the key aspect of the adsorption. The loading content on graphite is 10 wt.% (Figure 2(2)), roughly consistent with this hypothesis.

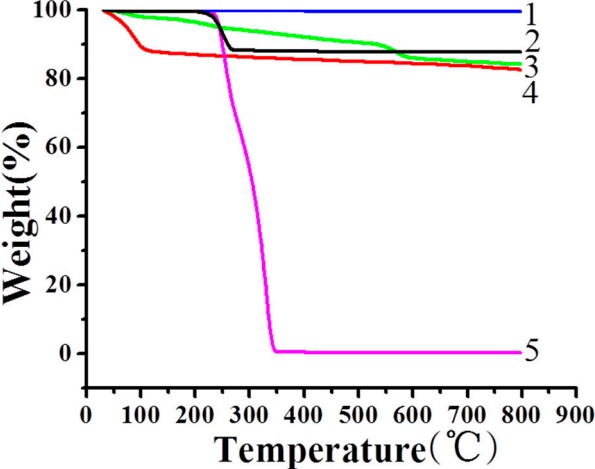

**Figure 2.** TGA of (1) graphite, (2) L-proline/graphite, (3) L-proline/activated carbon, (4) activated carbon, and (5) neat L-proline. The weight loss of L-proline was measured in its decomposition range of 229–347 °C.

To further clarify the loading pattern of L-proline on the MWCNTs, a series of control experiments were conducted. The loaded L-proline was not lost after 3 or 5 rounds of washing with water (Figure 1(d3,e7)). Washing with 10% aq. $H_2SO_4$ had no obvious effect (Figure 1(e8)), whereas washing with 10% aq. NaOH significantly reduced the L-proline content (Figure 1(e4)). It is possible that the carboxyl end of L-proline plays an important role in its adsorption and thus, deprotonation in alkaline solution would compromise its effects. Furthermore, the possibility of the chelation between L-proline and the metal impurities in the MWCNTs was ruled out, since there were only trace metals detected by XRF (X-Ray Fluorescence) and XPS (X-Ray Photoelectron Spectroscopy), in which the majors included 0.3% of Mg, 0.9% of Al and 0.3% of Co.

In FT-IR analysis (Figure S1, SM), the MWCNTs showed no distinct peaks of functional groups, whereas free L-proline showed characteristic peaks at 1627 (C=O) and 2769 $cm^{-1}$ (N–H). For the L-proline/MWCNTs catalyst, the spectrum displayed similar peaks to those of free L-proline, proving its successful loading on the MWCNTs. Moreover, the BET ($N_2$ adsorption and desorption curve, and the pore size distribution, see Figure S2 in SM) surface areas for the MWCNTs and L-proline/MWCNTs catalysts were determined to be 110.0215 $m^2/g$ and 37.4382 $m^2/g$, respectively. The remarkable reduced value further confirmed the loading of L-proline on the MWCNTs. In scanning electron microscopy (SEM, Figure S3, SM), the loading of L-proline had almost no influence on the morphology of MWCNTs.

## 2.2. Control Experiments

For comparison, zero-, one-, two-, and three-dimensional (0D, 1D, 2D and 3D) nanocarbons were studied as support for L-proline, namely $C_{60}$, single-wall CNTs (SWCNTs), GO, and activated carbon, respectively. Following similar synthetic procedure, the supported L-proline catalysts were obtained successfully. Their catalytic activity and stability in Aldol reaction were summarized in Table 1. It was found that L-proline/$C_{60}$ had almost no activity, possibly because that the spherical structure of $C_{60}$ makes it difficult for L-proline to adsorb. For SWCNTs, MWCNTs, and GO, the catalytic performance was all maintained (around 90% yield) after 7 rounds of reaction. Their common feature is the aromatic carbon rings and the oxygen-containing functional groups [38]. When loaded on the activated carbon, the catalyst exhibited good activity in the first round (93% yield), which quickly diminished to 5% after only 3 rounds of catalysis. As activated carbon contains lots of oxygen-containing functional groups and small cavities, its inability of retaining L-proline suggests that these factors by themselves are insufficient. Additional assistance by the aromatic carbon or hydrophobic interaction is probably needed.

**Table 1.** Catalytic activity and stability of L-proline/nanocarbons in Aldol reaction.

| Nanocarbon | Yield (%) [1] in 1st Round | Yield (%) [1] in (Rounds Number) |
|---|---|---|
| $C_{60}$ | trace | not available |
| Graphene oxide | 90 | 91 (7) |
| MWCNTs | 91 | 90 (7) |
| SWCNTs | 92 | 91 (7) |
| Activated carbon | 91 | 5 (3) |

Reaction conditions: The solution of L-proline/nanocarbons (29 mol% catalyst loading) and 4-nitrobenzaldehyde (0.151 g, 1.0 mmol) in acetone (5.0 mL) was stirred at room temperature for 6h under Ar. [1] High performance liquid chromatography (HPLC) yields.

Besides L-proline, the loading capacity of other small molecules similar to L-proline, more specifically glycine as a simple amino acid, arginine for its additional amine groups, cyclohexane for the aliphatic ring structure, and pyrrolidine for the similar moiety with L-proline were investigated. The obtained loading amount range of 0–43.9 wt.% for these molecules on the MWCNTs (Figure S4, SM) was much lower than that of L-proline. It was noted that carboxyl-free pyrrolidine and cyclohexane could not fix onto MWCNTs well, further confirming the importance of carboxylic group in the loading process. For larger molecules, such as quinine, the loading capacity on the MWCNTs was poor (Figure S4, SM).

Mutual exchange experiments were carried out using above small molecules. The MWCNTs were pre-treated with these molecules, and purified for 5 rounds before adsorbing L-proline. TGA showed that the L-proline loading was not significantly affected (59–67 wt.%, Figure 3). However, when the L-proline loaded MWCNTs were treated with aqueous emulsion containing excessive amount of the small molecules, more than half of the loaded L-proline was lost. In particular, cyclohexane/water emulsion removed 92% of the loaded L-proline, which is surprising considering that reaction in isopropanol/cyclohexane does not remove the L-proline (see the Michael reaction below). It is possible that the hydrophobic interaction was probably of importance in the L-proline adsorption.

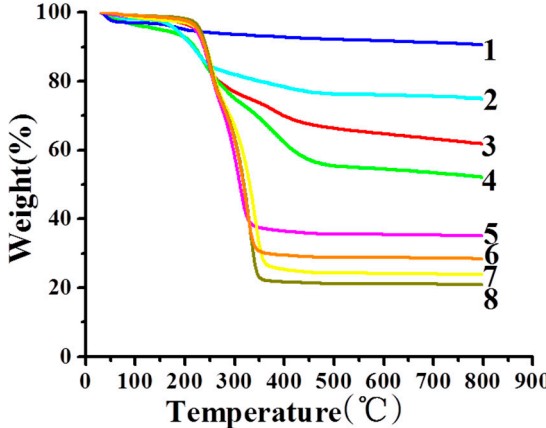

**Figure 3.** TGA of sample when (1) cyclohexane replacing L-proline, (2) pyrrolidine replacing L-proline, (3) glycine replacing L-proline, (4) arginine replacing L-proline, (5) L-proline replacing glycine, (6) L-proline replacing pyrrolidine, (7) L-proline replacing arginine, and (8) L-proline replacing cyclohexane. The weight loss of L-proline was measured in its decomposition range of 229–347 °C.

### 2.3. Application of the L-Proline/MWCNTs Catalysts

The catalytic activity and recyclability of the L-proline/MWCNTs catalyst were investigated in the classic Aldol reaction of acetone and 4-nitrobenzaldehyde (Figure 4). A mixture of L-proline/MWCNTs (0.050 g, containing 0.29 mmol of L-proline) and acetone (5.00 mL) was stirred at room temperature, followed by the addition of 4-nitrobenzaldehyde (0.151 g, 1.00 mmol). The amount of catalyst and reaction time (29 mol% for 8 h) are roughly consistent with those in the literature (10–30 mol% for 4–24 h) [39–41]. The upper organic phase was collected, concentrated, and then purified by the column chromatography (hexane/ethyl acetate, 2:1) to give the desired 4-hydroxy-4-(4-nitrophenyl)-butan-2-one as a pure compound. The yield of product was determined to be 91% (TON = 3.14) by HPLC and the catalytic activity was similar to the free L-proline (89% yield) under the same reaction conditions.

Temporal evolution of the reaction showed that the L-proline/MWCNTs catalyst was indeed faster than free L-proline in reaching the equilibrium at approximately 90% yield (Figure 5). It is important to note that this is achieved when the L-proline loading in the composite is extremely high, meaning that most of the loaded L-proline molecules are still active. Moreover, the enantioselectivity of the product was almost the same whether the L-proline/MWCNTs (ee: 68%) or the free L-proline (ee: 69%) was applied (Figure S5, SM). The supplementary Circular Dichroism spectra (Figure S6, SM) also confirmed the retention of the chirality induction of the L-proline before and after loading on the MWCNTs.

On the other hand, the residual solid catalyst after removing the supernatant was washed with acetone (5 mL × 3) and dried in oven, before it was re-applied in the consecutive rounds of reaction. The L-proline/MWCNTs maintained consistent catalytic efficiency even after 7 rounds of catalysis (Figure 4), i.e., the supported catalyst had not only high catalytic efficiency, but also excellent recycling stability.

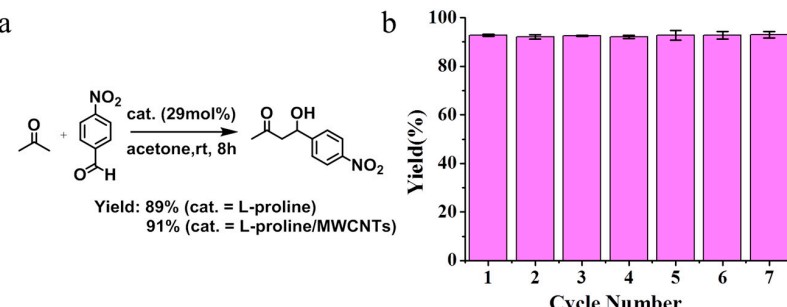

**Figure 4.** (**a**) Aldol reaction of acetone and 4-nitrobenzaldehyde catalyzed by L-proline or L-proline/MWCNTs and (**b**) recycling and reuse bar graph of L-proline/MWCNTs catalyst in Aldol reaction.

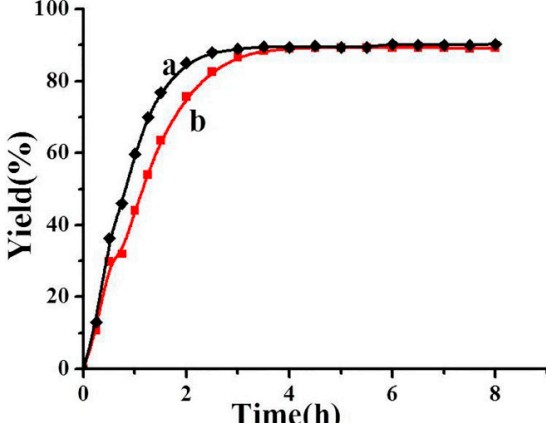

**Figure 5.** Temporal evolution of Aldol reaction catalyzed by (**a**) L-proline/MWCNTs and (**b**) L-proline.

Furthermore, the filtration test was conducted to investigate the possible leached L-proline in the reaction system. As shown in Figure S7 in SM, the HPLC analysis indicated that only 4% conversion yield of product was obtained when the L-proline/MWCNTs catalysts were filtered off from the reaction before adding the substrates. By the comparison with 90% conversion yield of the product with the L-proline/MWCNTs catalysts, the possible leached L-proline could be ignored.

Encouraged by the successful application in Aldol reaction, more reactions were tested to explore the general applicability of the supported catalyst (Figure 6). The detailed description of the reactions was included in SM. For Mannich reaction (Figure 6a) and Knoevenagel condensation (Figure 6d) in DMSO (dimethyl sulfoxide), the L-proline/MWCNTs exhibited excellent catalytic activity similar to the free L-proline with the product yield up to 99%. In a mixed solvent of isopropanol and cyclohexane (V/V = 1:4), Michael reaction (Figure 6b) catalyzed by the L-proline/MWCNTs gave the desired product with similar yield (75%) as the free L-proline (72%). $\alpha$-Oxyamination reaction (Figure 6c) in DMF (N,N-Dimethylformamide) was also investigated and ~90% yield was obtained whether the L-proline/MWCNTs catalyst or free L-proline was applied. For all the tested reactions above, the L-proline/MWCNTs catalyst retained the catalytic efficiency after 7 rounds of recycling and reuse (Figures S8–S11, SM), demonstrating excellent recycling stability in various reaction and solvent systems. Furthermore, in comparison with the literature results for Mannich reaction, Knoevenagel condensation and $\alpha$-Oxyamination reaction, it is found that the catalytic efficiency of the L-proline/MWCNTs is as good as the reported values [42–44]. For Michael reaction, our catalyst efficiency is slightly lower, but the recycling stability is much better [45].

**Figure 6.** Application of the L-proline/MWCNTs catalyst in (**a**) Mannich reaction, (**b**) Michael reaction, (**c**) α-oxyamination reaction, and (**d**) Knoevenagel condensation.

## 3. Materials and Methods

### 3.1. Materials

4-Nitrobenzaldehyde (99%, J & K, Beijing, China), benzaldehyde (98%, J & K, Beijing, China)), *p*-methoxyaniline (99%, Meryer, Shanghai, China) and cyclohexanone (analytical reagent (AR), Meryer, Shanghai, China) are used without further purification. *n*-Pentanal (98%), xylene malonate (99%), 2-nitrotoluene (AR), nitrosobenzene (AR) and L-proline (AR) are purchased from local suppliers (Wanqing Company, Nanjing, Jiangsu, China). Multi-wall carbon nanotubes (MWCNTs) and other carbon nanomaterials are obtained from XFNANO Ltd. (Nanjing, Jiangsu, China). All other reagents and solvents, unless otherwise noted, are purchased from commercial vendors (Wanqing Company, Nanjing, Jiangsu, China) and used without further purification.

### 3.2. Equipments

All reactions are monitored by thin-layer chromatography (TLC) performed on pre-coated TLC plates (silica gel, 0.2 mm, HSGF254, Wanqing Company, Nanjing, Jiangsu, China). Visualization is accomplished with ultraviolet (UV) light (254 nm) and phosphomolybdic acid in ethanol by heating. Flash column chromatography is performed on silica gel (200–300 mesh, Huanghai Ltd., Yantai, Shandong, China). $^1$H NMR and $^{13}$C NMR spectra are recorded on a JNM-ECZ400S spectrometer (JEOL, Tokyo, Japan). The TGA of ca. 15 mg of sample is performed on a METTLER TOLEDO thermogravimetry (METTLER, Zurich, Switzerland). Infrared spectrum is obtained through a FT-IR spectrometer (Thermo Fisher, Waltham, MA, USA). Elemental analysis is tested by Shanxi Kaiencheng Detection Technology Co, Ltd. (Xian, Shanxi, China). Conversion yield is measured by HPLC (ACQUITY Arc, Waters, Shanghai, China). The morphology of substance surface is determined on a scanning electron microscopy (FEI, Hillsboro, OR, USA).

### 3.3. Typical Procedure for Catalyst Preparation

The L-proline/MWCNTs catalyst was synthesized by mixing MWCNTs (0.10 g) and aqueous L-proline (1.50 g, 13.0 mmol) with stirring at room temperature. After 3 rounds of washing with water and centrifugal separation, the precipitate was collected and dried overnight under vacuum to give L-proline/MWCNTs as a black powder.

## 4. Conclusions

In conclusion, L-proline was successfully loaded on the MWCNTs by simple mixing, with exceptionally high loading content of up to 67 wt.%, which is much higher than previous literature-reported values. It is equivalent to one L-proline molecule per 3.2 carbon atoms, suggesting that many of them were trapped in the cavities of MWCNTs. TGA, elemental analysis, and catalytic performance jointly confirmed the abnormal high loading content. On the bases of extensive control experiments, we speculate that L-proline may adsorb via its carboxyl end assisted by hydrophobic interactions. The catalytic performance of the loaded L-proline is on par with free L-proline and even slightly better, again unusual for loaded organo-catalysts. Either the surface adsorbed L-proline has superior activity, or the reactants could reach inside the cavities of MWCNTs. The L-proline/MWCNTs catalyst is generally applicable for 5 typical L-proline catalyzed reactions, even in different solvent systems. Most importantly, the exceptional loading capacity of MWCNTs for L-proline and the surprising catalytic performance of the supported catalyst would offer insights into CNTs as a general support for organo-catalysts. Such a new window for future catalyst design may facilitate the practical application of organo-catalysts as large-scale recyclable catalyst.

**Supplementary Materials:** The following are available online at http://www.mdpi.com/2073-4344/10/11/1246/s1, Figure S1: Infrared spectra of (a) MWCNTs, (b) L-proline, and (c) L-proline/MWCNTs., Figure S2. (A) $N_2$ adsorption and desorption curves of MWCNTs (red trace) and L-proline/MWCNTs (blue trace), (B) the pore size distribution of MWCNTs (black trace) and L-proline/MWCNTs (red trace), Figure S3: SEM of (a) MWCNTs and (b) L-proline/MWCNTs., Figure S4. TGA of (1) MWCNTs, (2) cyclohexane/MWCNTs, (3) pyrrolidine/MWCNTs, (4) quinine/MWCNTs, (5) glycine/MWCNTs, (6) arginine/MWCNTs., Figure S5: Chiral HPLC of product when (a) L-proline/MWCNTs and (b) L-proline applied., Figure S6: Circular dichroism spectra of L-proline/MWCNTs (green) and neat L-proline (red). The solid samples were tested by potassium bromide tablet at room temperature and 999.985 V., Figure S7: HPLC of product without (a) and with (b) filtration treatment., Figure S8: Recycling and reuse of L-proline/MWCNTs catalyst in Mannich reaction., Figure S9: Recycling and reuse of L-proline/MWCNTs catalyst in Michael reactions., Figure S10: Recycling and reuse of L-proline/MWCNTs catalyst in $\alpha$-oxyamination reaction., Figure S11: Recycling and reuse of L-proline/MWCNTs catalyst in Knoevenagel condensation., Table S1: Elemental analysis of L-proline/MWCNTs.

**Author Contributions:** Conceptualization, J.X., H.C. and Y.Z.; experiment characterization and analysis, J.X., J.L. and S.H.; software, K.T., R.C. and G.Y.; supervision, Y.Z. All authors have read and agreed to the published version of the manuscript.

**Funding:** The authors thank National Natural Science Foundation of China (No. 21673117 and 91956109), Jiangsu Provincial Foundation for Specially Appointed Professor, and Start-up Fund from Nanjing Tech University (No. 39837126 and 39837102).

**Conflicts of Interest:** The authors declare no conflict of interest. The funders had no role in the design of the study; in the collection, analyses, or interpretation of data; in the writing of the manuscript, or in the decision to publish the results.

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
