# Peer review of "On the Exceptionally High Loading of L-Proline on Multi-Wall Carbon Nanotubes"

_catalysts, doi:10.3390/catal10111246_

Round 1
Reviewer 1 Report
Manuscript Number: catalysts-959900
Title: On the Exceptionally High Loading of L-proline on Multi-Wall Carbon Nanotubes
General Comments:
In this work, L-proline is directly loaded on multi-wall carbon nanotubes (MWCNTs) with exceptionally high loading content of 67 wt%. The obtained L-proline/MWCNTs catalyst was on par with the catalytic activity of free L-proline, even after 7 rounds of recycling and reusing. It was suggested that the excellent activity of L-proline/MWCNTs in typical Aldol reaction, Mannich reaction, Michael reaction, α-oxyamination reaction, and Knoevenagel condensation shows a broad applicability of the composite catalyst in different reactions and solvent systems. Furthermore, it can be summarized that the unusual loading mode may open a window for designing heterogenized organo-catalysts. The article is interesting; it has enough experimental work and adds some new information especially on providing the exceptionally high loading of L-proline on multi-wall carbon nanotubes. Moreover, it adheres to the journal’s standards as Catalysts is an international open access journal of catalysts and catalyzed reactions. Thus, my recommendation is to be accepted after major revision.
Specific Comments:
- English should be improved by a native speaker
- The innovation of the presented work should be clearly demonstrated
- A detailed explanation of the influence of the synthesis method - in correlation with their characteristic properties - to their catalytic performance should be provided.
- Some more works from the literature should be added and discussed through this manuscript.
Author Response
Referee1:
General Comments:
In this work, L-proline is directly loaded on multi-wall carbon nanotubes (MWCNTs) with exceptionally high loading content of 67 wt%. The obtained L-proline/MWCNTs catalyst was on par with the catalytic activity of free L-proline, even after 7 rounds of recycling and reusing. It was suggested that the excellent activity of L-proline/MWCNTs in typical Aldol reaction, Mannich reaction, Michael reaction, α-oxyamination reaction, and Knoevenagel condensation shows a broad applicability of the composite catalyst in different reactions and solvent systems. Furthermore, it can be summarized that the unusual loading mode may open a window for designing heterogenized organo-catalysts. The article is interesting; it has enough experimental work and adds some new information especially on providing the exceptionally high loading of L-proline on multi-wall carbon nanotubes. Moreover, it adheres to the journal’s standards as Catalysts is an international open access journal of catalysts and catalyzed reactions. Thus, my recommendation is to be accepted after major revision.
Specific Comments:
1.English should be improved by a native speaker
RESPONSE: We thank the reviewer for the comments. English has been modified according to the suggestions of a native speaker, as highlighted in the text.
2.The innovation of the presented work should be clearly demonstrated
RESPONSE: We thank the reviewer for the helpful comments. The innovation of this work has been emphasized in the conclusion part (page 8, line 242-244). L-proline was successfully loaded on the MWCNTs by simple mixing, with exceptionally high loading content of up to 67 wt%, which is much higher than previous literatures reported values. The exceptional loading capacity of MWCNTs for L-proline and the surprising catalytic performance of the supported catalyst would offer insights into CNTs as a general support for organo-catalysts.
3.A detailed explanation of the influence of the synthesis method - in correlation with their characteristic properties - to their catalytic performance should be provided.
RESPONSE: Thanks for the reviewer’s comments. In the literatures, covalently bonding are usually used to realize the loading of the organic catalysts. Although the loaded organo-catalysts can be recycled and reused, their reactivity and selectivity are often significantly lower than that of the free organic catalysts [ref. 13]. This may be due to the heterogeneity of the homogeneous organic catalyst. The discussion has been added to the main text on page 1, line 33-35.
In addition, on page 2, line 44-46, we cite a new reference to show the influence of the synthesis method. Chandrasekarna et al. report that L-proline could be covalently fixed on the MCM-41 (a silicon-based support), but the loading amount is only about 28 wt%, and the catalytic efficiency is poor [ref. 23].
4.Some more works from the literature should be added and discussed through this manuscript.
RESPONSE: We thank the reviewer for the helpful suggestion. More relevant references have been added in the main text, including ref. 13, 14, 22, 28, 29, 30, 34, 35, 36, 42, 43, 44, and 45. The serial numbers of other references are adjusted accordingly.
On page 2, line 54-58, we cite ref. 28 and ref. 29 about the covalent functionalization of the MWCNTs with the proline derivatives, with the maximum catalyst loading of 15 wt% and 7 wt%, respectively. When the catalysts are applied in Aldol reaction or Mannich reaction, although the yield of the product is up to 95%, the catalytic efficiency decreases dramatically after only 3 or 5 rounds of recycling.
Reviewer 2 Report
In my opinion, an exceptionally nice paper, very complete, with several reactions being studied, with very good results and recyclbility and a very high loading of catalyst on the nanotubes. I just authors to please confirm and state on the Introduction, what was the maximum loading of proline on CNTs reported in literature before this paper and in what reactions it had been studied.
Also compare the results obtained for these reactions with others from literature for CNT catalysts (or others) so that we can also have a better idea on the efficiency of the catalyst.
Finally, some minor remarks:
Figure 1 is confusing. CNT, proline and CNT with proline drawings, although very nice, are marked with a, b) c), which is confusing with the curves from the graphics below, as the following graphics should ne d) and e). Even better, graphics should be named a) and b), and the curves should have a different nomenclature, like 1,2,3, that would also correspond to the structures. Similar nomenclature should be given to the curves of Figure 2. Figure 3 has the right nomenclature, althoigh 1,2,3, should be sequential, te ower numbers should correspond to the top curves and now there is a random distribution. The same with Figure S4.
Figure S2 should be complemented with the pore size distribution as well.
Table 1 does not have a caption, instead the text of the template is there.
Round 2
Reviewer 2 Report
In my opinion, the authors replied well to the referee comments and the paper improved substantially. So it can now be accepted for publication.